# A ViSAR Shadow-Detection Algorithm Based on LRSD Combined Trajectory Region Extraction

**Zhongzheng Yin** [1,2,*], **Mingjie Zheng** [1] **and Yuwei Ren** [1,2]

1   Aerospace Information Research Institute, Chinese Academy of Sciences, Beijing 100190, China
2   School of Electronic, Electrical and Communication Engineering, University of Chinese Academy of Sciences, Beijing 100049, China
*   Correspondence: yinzhongzheng20@mails.ucas.ac.cn

**Abstract:** Shadow detection is a new method for video synthetic aperture radar moving target indication (ViSAR-GMTI). The shadow formed by the target occlusion will reflect its real position, preventing the defocusing or offset of the moving target from making it difficult to identify the target during imaging. To achieve high-precision shadow detection, this paper proposes a video SAR moving target shadow-detection algorithm based on low-rank sparse decomposition combined with trajectory area extraction. Based on the low-rank sparse decomposition (LRSD) model, the algorithm creates a new decomposition framework combined with total variation (TV) regularization and coherence suppression items to improve the decomposition effect, and a global constraint is constructed to suppress interference using feature operators. In addition, it cooperates with the double threshold trajectory segmentation and error trajectory elimination method to further improve the detection performance. Finally, an experiment was carried out based on the video SAR data released by Sandia National Laboratory (SNL); the results prove the effectiveness of the proposed method, and the detection performance of the method is proved by comparative experiments.

**Keywords:** video SAR; moving target detection; shadow detection; LRSD

## 1. Introduction

Video synthetic aperture radar [1] is a high-resolution, high-frame-rate radar imaging system. It can focus on a specific area and present information in the two dimensions of time and space, so it has significant advantages in areas such as regional dynamic monitoring and change detection [2,3]. ViSAR has a strong ability to detect moving targets. Since SNL proposed the concept of ViSAR in 2003 and achieved high-frame-rate imaging using an airborne platform, it has attracted widespread attention at home and abroad.

Moving target detection is an important research direction in the field of SAR. It can be divided into single-channel and multi-channel moving target detection methods. Doppler frequency shift and the difference in modulation frequency are the main bases for the single-channel method to distinguish moving and static targets. The filters are designed to filter out clutter in the frequency domain, such as frequency detection and pre-filtering methods. The core idea of the multi-channel detection method is clutter suppression, which eliminates clutter and retains moving targets to achieve detection. Commonly used methods include displaced phase center antenna (DPCA), along-track interference (ATI), and space-time adaptive processing (STAP) [4–6]. The traditional SAR-GMTI technology mainly focuses on the backscattering energy of the target and has high requirements for the signal–clutter–noise ratio (SCNR). The single-channel detection method is effective for moving targets that deviate from the clutter spectrum, but it cannot detect targets submerged in the clutter spectrum, and the minimum detectable speed is relatively large [7]. The video SAR system has a shorter synthetic aperture time, which makes it easy to leave target shadows in the image. On the one hand, the true position of the moving target can be accurately found

through shadow detection, and on the other hand, the detection of the moving target no longer depends on the radar cross-section (RCS) of the target itself. Therefore, shadow detection is another important method of video SAR moving target detection.

In recent years, many scholars have studied the detection algorithm for moving target shadows. Classical image processing methods are used in [8–10] for video SAR image processing, including image denoising and image segmentation to construct the background, and the background difference method is used in the detection of moving target shadows. However, the classic image processing theory has strong applicability in simple scenes but does not have high detection accuracy in complex scenes, so a large number of false alarms and missed detections occur. Deep learning networks are used in [11–13] to detect moving target shadows in video SAR. The authors of [13] added a Feature Pyramid Network (FPN) to the model to strengthen the network's feature-extraction ability. An attention mechanism is used in [14] to focus on the key information to improve detection performance. However, deep learning networks do not show strong performance when faced with object-detection tasks from different data sources. The method of matrix decomposition is used in [15] for moving object detection. The authors of [16] used the robust principal component analysis (RPCA) model for video SAR moving target detection. In this paper, the parameter setting of LRSD was optimized, the performance of matrix decomposition was improved, and a new detector was proposed based on the foreground image with an unknown statistical distribution. Ideally, the foreground (moving target) and background in the video frame can be accurately separated based on the low-rank sparse decomposition method. However, the actual situation is that the matrix decomposition result is susceptible to system noise and complex interference from the environment, resulting in misclassification and omission. Each pixel is regarded as an independent point in the RPCA model, which does not consider the specific spatial structure and time information of the target and has poor robustness in the face of complex scenes with dynamic backgrounds and strong clutter interference.

This paper proposes a new video SAR moving target shadow-detection method to deal with the problems discussed above. Based on the idea of low-rank sparse decomposition, this method designs a new decomposition framework to improve the performance of matrix decomposition. The new model incorporates coherent expression items and total variation-based regularization and uses the space–time information of the target and clutter to improve the distinction between the two. The feature-extraction operator is used to construct the mapping relationship and provides support for the subsequent high detection accuracy. On the other hand, this paper adopts the scheme of dual-threshold trajectory-area extraction combined with error trajectory-area elimination. By performing double-threshold determination on the foreground image obtained by decomposition, the trajectory of the moving target is retained, and the interference area is eliminated according to the spatiotemporal characteristics of the moving target and the clutter difference. This process effectively suppresses clutter to achieve reliable moving target detection.

This paper is structured as follows. The improved LRSD model and target trajectory area extraction method are introduced and the theoretical introduction and solution steps are explained in Section 2. The relevant parameters of the proposed method and the experimental results are given in Section 3. An ablation experiment for the improved LRSD model was carried out to prove the effectiveness of the improved model, and then comparative experiments with other methods were set up to prove the improvement in the detection efficiency using the proposed method. Section 4 analyzes the results of the proposed method from the perspective of previous research, and at the same time provides an outlook on future research and development. Section 5 summarizes the applicability and advantages of the method.

## 2. Methodology

In this section, the RPCA theory is introduced in Section 2.1, and the applicability and limitations of shadow detection are analyzed. Then, based on these analyses, a solution is

proposed in Section 2.2, and the algorithm flowchart of this solution is shown in Figure 1. The solution steps of the algorithm are given in Section 2.3.

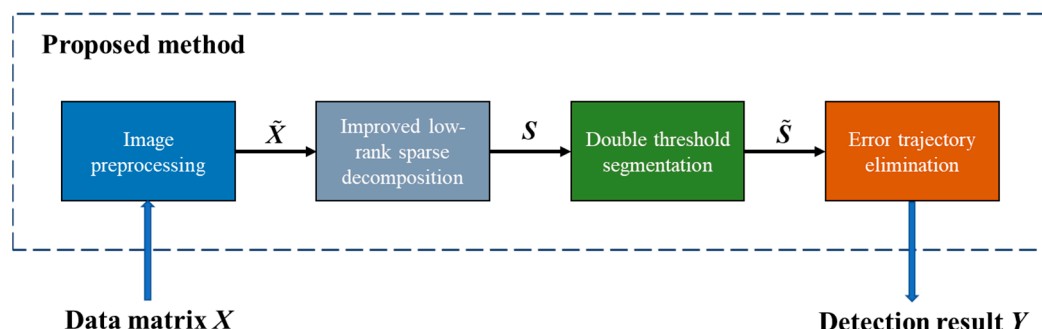

**Figure 1.** Algorithm flowchart.

### 2.1. RPCA Theory of Video SAR

Background modeling is a classic method of moving object detection. It includes median modeling, Gaussian Mixture Model (GMM), visual background extractor (VIBE), etc. In recent years, a method based on matrix decomposition has been proposed and widely used in background modeling, image denoising, and object detection. The most famous is the RPCA model proposed by Wright [15], which considers that the background part of the video image can be captured by a low-rank matrix. At its core, it is a sparse and low-rank representation of matrices. Moving objects are included in the foreground matrix, and background areas are included in the background matrix. The target can be obtained by extracting the foreground matrix. With this method, $p$ frames of video images with $m \times n$ pixels are used to construct an observation matrix $D$ with $m \times n$ rows and $p$ columns. The observation matrix $D$ can be decomposed into a low-rank matrix $L$ representing the background and a sparse matrix $S$ of foreground moving objects. Furthermore, the convex approximation of the $L_0$ norm and rank function is made by using the $L_1$ norm and nuclear norm, respectively, and the RPCA model after convex relaxation can be expressed by Equation (1).

$$\min_{L,S} \quad \|L\|_* + \lambda \|S\|_1$$
$$\text{s.t.} \quad D = L + S \tag{1}$$

where $\|\cdot\|_*$ is the nuclear norm of the matrix, which is the sum of the eigenvalues of the matrix, and $\|\cdot\|_1$ is the norm of the matrix. $\lambda$ is the weight parameter. The inexact Lagrange multiplier method (IALM) and alternating direction multiplier method (ADMM) can be used to solve the above optimization problems.

There are also image sequences in video SAR. In the circular or spotlight mode, the background in the video SAR image sequence is highly correlated, and such static objects can be considered to have low-rank characteristics. Moving targets occupy fewer pixels in SAR images. In shadow detection, most of the gray values of moving targets have a certain degree of distinction from the scene, so moving targets can be considered sparse. The observation matrix is constructed, and the RPCA model is used for matrix decomposition.

The video SAR image sequence was decomposed using the RPCA model based on the video SAR imaging data published by SNL [17] and is shown in Figure 2.

The real target position is marked with a blue box in Figure 2. There is a lot of interference (the road edge, strong scattering points, and dynamic background) in the foreground image of Figure 2c, and the foreground image cannot accurately extract the moving target. There are three main reasons for the poor decomposition effect. First, the model itself does not distinguish the moving target from the dynamic background, so it will attribute the target and noise points to a sparse matrix, because the $L_1$ norm cannot correctly provide a sparse measure of the moving target and the dynamic background. The second is that the spatio-temporal structure of the target is not considered in the model,

and the robustness is poor in complex scenes. The third is that not only do moving targets have shadows in the video, but the shadows of other objects also exist at the same time, which changes with the observation angle of the SAR system. There is a strong interference for detection. Most of the existing video SAR moving target detection methods only apply the RPCA model and do not design a new decomposition framework based on complex scenes to improve the decomposition performance.

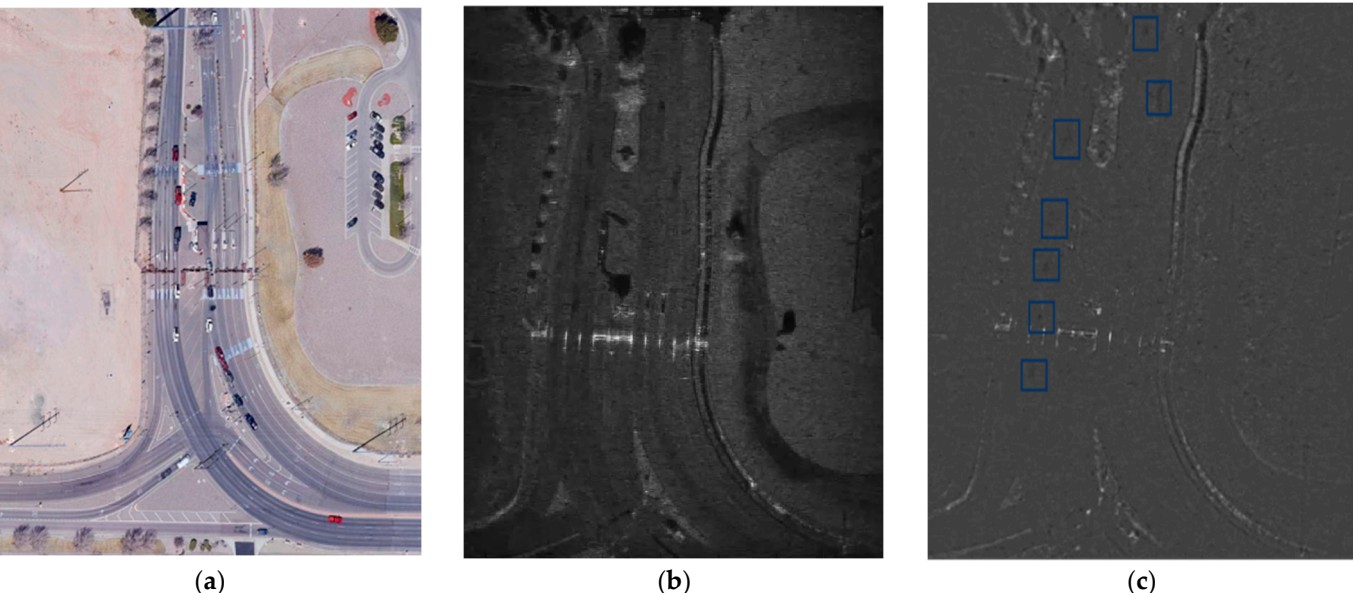

**Figure 2.** SNL video SAR data and RPCA decomposition results. (**a**) Actual scene diagram. (**b**) Video SAR imaging results. (**c**) RPCA decomposition results.

*2.2. Improved LRSD Model*

The RPCA model has many problems in actual moving target shadow detection. To better adapt to this detection task, this section improves on the weak points of the model's performance. The improvement plan consists of three parts. (1) Introduce total variation-based regularization to constrain the foreground target. (2) Add a coherent suppression item and dynamic background constraint item to build a new decomposition framework. (3) Apply a Sobel edge extraction operator and design global constraints to eliminate edge influence. This section introduces the principles of the model in detail, and the solution of the model is introduced in detail in Section 2.3.

(1)　The moving target shadow occupies a certain number of pixels, presents a low gray-level distribution, and shows spatial and temporal continuous characteristics in the image. The $L_1$ norm in the RPCA model provides a relatively broad description of the low-rank characteristics of moving targets, but the noise and clutter in the image also have low-rank properties, which will seriously affect the matrix decomposition results. Considering the shadow-distribution characteristics of moving objects and the relative-motion characteristics between frames, a weighted total variation is used instead of the $L_1$ norm to constrain the foreground matrix. In other applications, the total variation function will bring a certain degree of smoothness. Here, it is assumed that the dynamic background is sparser than the smoothed foreground [18,19]. Objects that transform smoothly and have few sharp edges will have a low TV value, while sparse damage will have a very high TV value, so the total variation can be used as a sparse measure of the foreground target and the dynamic background. Equation (2) is a TV expression.

$$TV(\boldsymbol{X}) = \sum_{ijk}\left(w_{ijk}^x\left|x_{i+1jk} - x_{ijk}\right| + w_{ijk}^y\left|x_{ij+1k} - x_{ijk}\right| + w_{ijk}^z\left|x_{ijk+1} - x_{ijk}\right|\right) \qquad (2)$$

In Equation (2), $x = vec(X)$, $X \in R^{mnp}$, $w_{ijk}^v$ represents the weight on the vector $\vec{v}$, and $x_{ijk}$ represents the $(x, y)$ point of the $p$-th frame image. The following new LRSD model can be proposed.

$$\min_{L,S,E} \quad \|L\|_* + \lambda_s \mathrm{TV}(S) + \lambda_E \|E\|_1 + \frac{1}{2}\|N\|_F^2$$
$$\text{s.t.} \qquad D = L + S + E + N \qquad (3)$$

The foreground target constraints are changed and a dynamic background term $E$ is added to the matrix decomposition term in Equation (3). We do not want to divide the moving target and the dynamic background into one place in the original RPCA model, so we set a new decomposition item to receive the dynamic background part while optimizing the foreground matrix constraints.

It is practical to introduce the TV-constrained foreground matrix in the RPCA model. In the optical images, the improved model using the TV penalty instead of the $L_1$ norm can significantly improve the detection of moving objects. However, there are many differences in the imaging mechanism and characteristics of SAR images and optical images, and the TV penalty cannot have as good an effect in SAR images as in optical images. The following steps continue to solve this problem.

(2) The applicability of the total variation is restricted by the assumptions, and the decomposition effect of the model decreases when the assumptions are incorrect. A new decomposition framework is constructed by introducing the dynamic background constraint items and correlation-suppression items. The former is used to independently divide the dynamic background space, and the latter makes the moving target and the dynamic background better distinguishable. The new model is as follows.

$$\min_{L,S,E} \quad \|L\|_* + \lambda_s TV(S) + \lambda_E \|E\|_1 + \lambda_R R(S,E) + \frac{1}{2}\|N\|_F^2$$
$$\text{s.t.} \qquad D = L + S + E + N \qquad (4)$$

where $R(S,E)$ is the coherence suppression term between the constructed sparse foreground and dynamic background. $R(S,E) = \sum_{i=1}^{p} S_i^T E_i$, $S \in R^{m \times n \times p}$, $E \in R^{m \times n \times p}$, and $S_i$ and $E_i$ represent the $i$ th column of $S$ and $E$. $D$ ($D \in R^{m \times n \times p}$) is the observation matrix composed of video frames. $N$ ($N \in R^{m \times n \times p}$) is the noise term obtained by decomposition. In particular, the above matrices are all in a two-dimensional format with a size of $(m \times n) \times p$. $\|\cdot\|_F^2$ is the Frobenius norm. The Frobenius norm-constrained noise term comes from the related literature that proves that SAR image noise can be described by approximate Gaussian noise [20]. In particular, coherence suppression is essentially the introduction of image space–time information. Introducing more reasonable prior information into the model is a way to improve the model, and the correlation operation is just one of the expressions. The reason why we use this method is that, on the one hand, it has a straightforward expression and is easy to understand. On the other hand, the related operations are related to the trace and the Frobenius norm, which is convenient for solving optimization problems.

(3) Due to the problem that the road edge is seriously affected by the actual decomposition effect, the Sobel edge operator is used to extract these interference areas. A global constraint composed of a mapping function is added to the model to eliminate the influence of this interference. The Sobel operator is a commonly used edge detection method. It is essentially based on the convolution of the image space domain and is supported by the theory of the first derivative operator of the image. This method has a fast processing speed and has a smoothing effect on noise. The extracted edges are smooth and continuous, making them more suitable for this extraction task. The final model is presented as follows.

$$\min_{L,S,E} \quad \|L\|_* + \lambda_s TV(S) + \lambda_E \|E\|_1 + \lambda_R R(S,E) + \frac{1}{2}\|N\|_F^2$$
$$\text{s.t.} \qquad F(D) = F(L + S + E + N) \qquad (5)$$

where $F(\cdot)$ is a mapping function obtained after the Sobel operator [21] processes video frames. The process can be described as follows. First, the Sobel operator is used to extract edge features from the superposition results of multi-frame video SAR images and construct an edge feature mask $\boldsymbol{M}$, and then it constructs the following mapping relationship for the video frame $\boldsymbol{P}$.

$$F(\boldsymbol{X}(x,y)) = \begin{cases} \boldsymbol{X}(x,y), M(x,y) = 0 \\ 0, M(x,y) = 1 \end{cases} \tag{6}$$

$\boldsymbol{X}(x,y)$ is the gray value of each frame. $M(x,y)$ is the feature mask value of the point $(x,y)$, $\boldsymbol{M} \in \mathrm{R}^{m \times n}$. This process can be described in Figure 3.

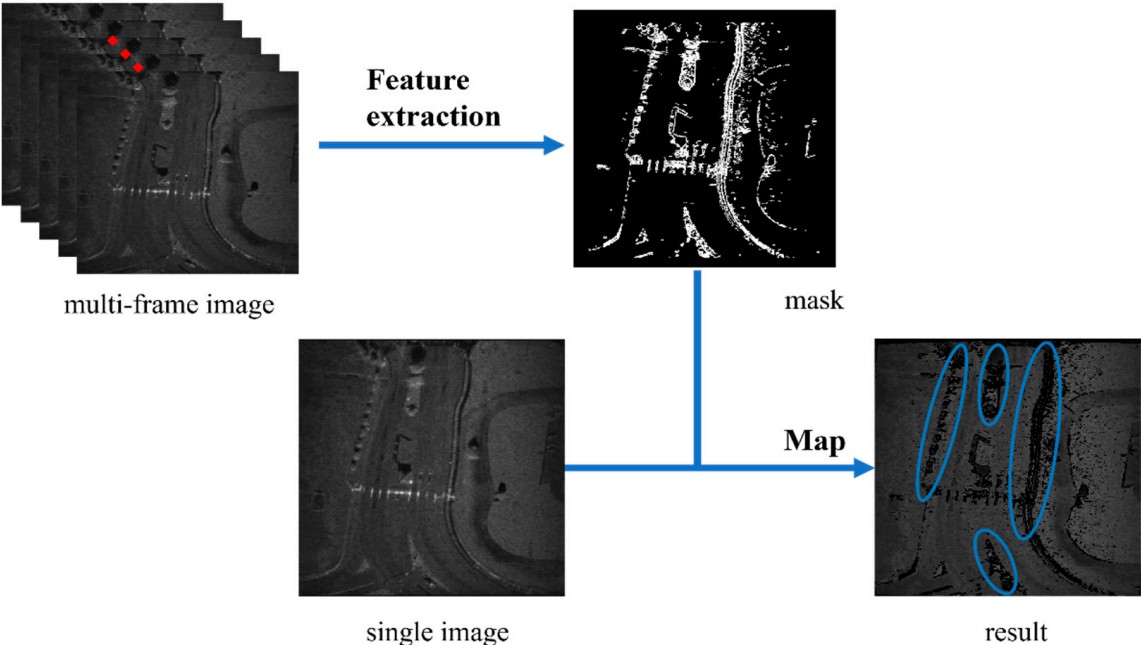

**Figure 3.** The process of building a mapping function.

### 2.3. Solution of the New Model

The new matrix decomposition model is given in the previous section. In this section, we mainly discuss its solution. First, the model is simplified into an unconstrained problem.

$$\min_{L,S,E} \quad \|\boldsymbol{L}\|_* + \lambda_s TV(\boldsymbol{S}) + \lambda_E \|\boldsymbol{E}\|_1 + \lambda_R R(\boldsymbol{S},\boldsymbol{E}) + \frac{1}{2}\|\boldsymbol{D}-\boldsymbol{L}-\boldsymbol{S}-\boldsymbol{E}\|_F^2 \tag{7}$$

The model is solved using the proximal gradient method [22]. With this method, it is assumed that the objective function can be decomposed into two parts, $f(z) = g(z) + h(z)$, where $g(z)$ is a differentiable convex function and $h(z)$ is a convex function. The proximal gradient operator gives the following expression.

$$\mathrm{prox}_h(x) := \underset{z}{\mathrm{argmin}} \frac{1}{2}\|z-x\|_F^2 + h(z) \tag{8}$$

Its iterative form is shown in Equation (8).

$$x^{k+1} = \mathrm{prox}_{h,\tau}(x^k - \tau^k \nabla g(x^k)) \tag{9}$$

In Equation (9), $\nabla$ is the gradient operator, and $\tau$ is the step size. The convergence condition of the proximal gradient method is given in [23]. The range of the convergence step size for similar problems is given in [24] for setting the reference step size.

The optimization problem of Equation (7) is split. The functions $g(\boldsymbol{L},\boldsymbol{S},\boldsymbol{E}) = \lambda_R R(\boldsymbol{S},\boldsymbol{E}) + \frac{1}{2}\|\boldsymbol{D}-\boldsymbol{L}-\boldsymbol{S}-\boldsymbol{E}\|_F^2$ and $h(\boldsymbol{L},\boldsymbol{S},\boldsymbol{E}) = \|\boldsymbol{L}\|_* + \lambda_s TV(\boldsymbol{S}) + \lambda_E \|\boldsymbol{E}\|_1$ are defined, and the two parts,

after splitting, correspond to the proximal gradient method. We iterate over **L**, **S**, and **E** using the proximal gradient operator.

i.　　Update $\boldsymbol{L}^k$.

$$
\begin{aligned}
\boldsymbol{L}^{k+1} &= \underset{L}{\arg\min} \|\boldsymbol{L}^k\|_* + \tfrac{1}{2}\|\boldsymbol{D} - \boldsymbol{S}^k - \boldsymbol{E}^k - \boldsymbol{L}^k\|_F^2 \\
&\Leftrightarrow \text{prox}_{\|\cdot\|_*,\tau^k}\left[\boldsymbol{L}^k - \tau^k\nabla_L g(\boldsymbol{L}^k, \boldsymbol{S}^k, \boldsymbol{E}^k)\right] \\
&= \text{prox}_{\|\cdot\|_*,\tau^k}\left[\boldsymbol{L}^k - \tau^k(\boldsymbol{L}^k + \boldsymbol{S}^k + \boldsymbol{E}^k - \boldsymbol{D})\right]
\end{aligned}
\tag{10}
$$

The conventional way to solve Equation (10) is to use the singular value-shrinkage operator [25], where the fixed threshold shrinkage will lose the energy of the effective part of the image. A superior low-rank solver OptShrink is proved in [26], which is used in [24] to generate better low-rank components. The definition of the solver is given below.

$$
\text{OptShrink}_\gamma(\boldsymbol{X}) = \sum_{i=1}^{\gamma}\left(-2\frac{D_{\mu X}(\sigma_i)}{D'_{\mu X}(\sigma_i)}\right)u_i v_i^H
\tag{11}
$$

where $\boldsymbol{X} = \sum_i \sigma_i u_i v_i^T$ is the singular value decomposition, and $\boldsymbol{X} \in \mathrm{R}^{m\times n}$. A variation is defined whose expression is $D_\mu(x) = \int \frac{x}{x^2-t^2}d\mu(t) \times \left[c\int \frac{x}{x^2-t^2}d\mu(t) + \frac{1-c}{x}\right]$. $D'_\mu(x)$ is the derivative of $D_\mu(x)$. $c$ is a constant. $c = \frac{\min(m,n)}{\max(m,n)}$. $\mu X(t) = \frac{1}{q-\gamma}\sum_{i=\gamma+1}^{q}\delta(t-\sigma_i)$ is an empirical function, and $q = \min(m,n)$. In particular, the $\min(\cdot)$ operator means taking the minimum value operation. Therefore, Equation (10) can be updated as shown in Equation (12).

$$
\boldsymbol{L}^{k+1} = \text{Optshrink}_\gamma[\boldsymbol{L}^k - \tau^k(\boldsymbol{L}^k + \boldsymbol{S}^k + \boldsymbol{E}^k - \boldsymbol{D})]
\tag{12}
$$

ii　　Update $\boldsymbol{S}^k$.

$$
\begin{aligned}
\boldsymbol{S}^{k+1} &= \underset{S}{\arg\min}\,\lambda_s TV(\boldsymbol{S}^k) + \lambda_R R(\boldsymbol{S}^k, \boldsymbol{E}^k) + \tfrac{1}{2}\|\boldsymbol{D} - \boldsymbol{L}^{k+1} - \boldsymbol{S}^k - \boldsymbol{E}^k\|_F^2 \\
&= \underset{S}{\arg\min}\,\lambda_s TV(\boldsymbol{S}^k) + \tfrac{1}{2}\|\boldsymbol{D} - \boldsymbol{L}^{k+1} - (1+\lambda_R)\boldsymbol{E}^k - \boldsymbol{S}^k\|_F^2 \\
&\Leftrightarrow \underset{S}{\arg\min}\,\lambda_s TV(\boldsymbol{S}^k) + \tfrac{1}{2}\|\boldsymbol{Z} - \boldsymbol{S}^k\|_F^2
\end{aligned}
\tag{13}
$$

The solution to the optimization problem of $S$ cannot be obtained directly, and it is necessary to convert the total variation into a computable form before solving it. The transformation matrix $\boldsymbol{D}_n(\boldsymbol{D}_n \in \mathrm{R}^{n\times n})$ is introduced to make it easier to solve the optimization problems.

$$
\boldsymbol{D}_n = \begin{bmatrix} -1 & 1 & & & & \\ & -1 & 1 & & & \\ & & \ddots & \ddots & & \\ & & & \ddots & \ddots & \\ & & & & -1 & 1 \\ 1 & & & & & -1 \end{bmatrix}
\tag{14}
$$

$\boldsymbol{D}_n$ is the difference matrix. $\boldsymbol{D}_n\boldsymbol{v}$ is a cyclic difference between adjacent elements of $\boldsymbol{v}$ where $\boldsymbol{v}$ is a vector. This operation rule fits the difference term along a certain dimension in the total variation. Therefore, in the case of a one-dimensional difference, $TV(v) = \|\boldsymbol{W}\boldsymbol{D}_n v\|_1$. For the three-dimensional case, the transformation matrix $\boldsymbol{C}$ is defined as follows.

$$
\boldsymbol{C} = \begin{bmatrix} \boldsymbol{I}_p \otimes \boldsymbol{I}_n \otimes \boldsymbol{D}_m \\ \boldsymbol{I}_p \otimes \boldsymbol{D}_n \otimes \boldsymbol{I}_m \\ \boldsymbol{D}_p \otimes \boldsymbol{I}_n \otimes \boldsymbol{I}_m \end{bmatrix}
\tag{15}
$$

In Equation (15), $I_j$ is a $j \times j$ identity matrix. Consider the closed solution of the penalty-function optimization problem, where $W$ is set as the identity matrix. Then, the total variational form in three dimensions can be given.

$$TV(X) = \|Cx\|_1 \tag{16}$$

From Equation (16), Equation (13) can be shown to be equivalent to the following vector expression.

$$\min_{S} \quad \lambda_s \|Cs\|_1 + \frac{1}{2}\|z - s\|_2^2 \tag{17}$$

In Equation (17), $z = \text{vec}(Z)$. When dealing with the optimization problem in Equation (17), some computational techniques are used to facilitate the solution. Transformation is defined as $u = Cs$. Equation (17) is constructed as an optimization problem with constraints, and ADMM [27] is used to solve it. A new optimization problem is proposed as follows.

$$\min_{s,v} \quad \lambda_s \|u\|_1 + \frac{1}{2}\|z - s\|_2^2 \\ \text{s.t. } Cs - u = 0 \tag{18}$$

Equation (18) is solved using the IALM and the ADMM method. The solution steps are given as follows.

$$s^{k+1} = \underset{s}{\text{argmin}}\frac{1}{2}\|z - s\|_2^2 + \frac{\mu}{2}\|Cs - u^k + Y^k\|_2^2 \\ u^{k+1} = \underset{u}{\text{argmin}}\lambda_s \|u\|_1 + \frac{\mu}{2}\|Cs^{k+1} - u + Y^k\|_2^2 \\ Y^{k+1} = Y^k + Cs - u^{k+1} \tag{19}$$

The objective function of the optimization problem of $s^{k+1}$ is differentiable and can be solved using the least squares method using the soft threshold shrinkage operator to update $u^{k+1}$. At the same time, the soft threshold shrinkage operator [28] is defined as $\text{soft}_\lambda(z) = \text{sign}(z)(|z| - \lambda)_+$, where $\text{sign}(\cdot)$ is the sign function. The solution result of Equation (18) can be obtained.

$$(I + \mu C^T C)s^{k+1} = z + \mu C^T(u^k - Y^k) \\ u^{k+1} = \text{soft}_{\frac{\lambda_s}{\mu}}(Cs^{k+1} + Y^k) \tag{20}$$

A new method is defined as $\text{TVM}_\lambda(Z) := \underset{s}{\text{argmin}}\frac{1}{2}\|Z - S\|_F^2 + \lambda_s TV(S)$. This formula is used to express the solution process of Equation (18) to Equation (20). The number of iterations is set to $K$. Equation (13) can be expressed as follows.

$$S^{k+1} = \text{prox}_{TV(\cdot),\tau^k\lambda_S}[S^k - \tau^k\nabla_S g(L^{k+1}, S^k, E^k)] \\ = \text{TVM}_{\lambda_S}[S^k - \tau^k(L^{k+1} + (1 + \lambda_R)E^k + S^k - D)] \tag{21}$$

iii     Update $E^k$.

$$E^{k+1} = \underset{E}{\text{argmin}}\lambda_E\|E^k\|_1 + \frac{1}{2}\|D - S^{k+1} - L^{k+1} - E^k\|_F^2 \\ \Leftrightarrow \text{prox}_{\|\cdot\|_1,\tau^k\lambda_E}[E^k - \tau^k\nabla_E g(L^{k+1}, S^{k+1}, E^k)] \\ = \text{prox}_{\|\cdot\|_1,\tau^k\lambda_E}[E^k - \tau^k(L^{k+1} + (1 + \lambda_R)S^{k+1} + E^k - D)] \tag{22}$$

Equation (22) can be solved by the soft-threshold-shrinkage operator mentioned above. Therefore, the optimization iteration problem of $E$ can be easily solved.

$$E^{k+1} = \text{soft}_{\tau^k\lambda_E}\left[E^k - \tau^k(L^{k+1} + (1 + \lambda_R)S^{k+1} + E^k - D)\right] \tag{23}$$

Based on the above derivation, the proposed problem can be simplified to Equation (24), and the algorithm solution process is given.

$$
\begin{aligned}
\boldsymbol{L}^{k+1} &= \text{OptShrink}_{\gamma}[\boldsymbol{L}^k - \tau^k(\boldsymbol{L}^k + \boldsymbol{S}^k + \boldsymbol{E}^k - \boldsymbol{D})] \\
\boldsymbol{S}^{k+1} &= \text{TVM}_{\lambda_S}[\boldsymbol{S}^k - \tau^k(\boldsymbol{L}^{k+1} + (1+\lambda_R)\boldsymbol{E}^k + \boldsymbol{S}^k - \boldsymbol{D})] \\
\boldsymbol{E}^{k+1} &= \text{soft}_{\tau^k \lambda_E}\left[\boldsymbol{E}^k - \tau^k(\boldsymbol{L}^{k+1} + (1+\lambda_R)\boldsymbol{S}^{k+1} + \boldsymbol{E}^k - \boldsymbol{D})\right]
\end{aligned}
\tag{24}
$$

The above-mentioned update iteration process involves basic matrix-derivation operations and form-conversion operations between traces and norms, which can be understood by consulting the relevant information. Many parameters need to be set during the iteration process. According to the parameter-setting theory in the RPCA model, the parameter setting before the decomposition item is $\lambda = \sqrt{\max(m, n)}$ [29,30], but in practice, it needs to be adjusted according to the size of the image and the decomposition effect. See Section 3 for specific parameter settings. The solution steps of the proposed model are shown in the Algorithm 1.

---

**Algorithm 1.** Proposed method

---

**Input:** Video frames $F_1, \cdots, F_p$
Parameters $\gamma > 0$, $\lambda_S > 0$, $\lambda_E > 0$, $\lambda_R > 0$
Iterations $K > 0$, parameters $\tau > 0$, $\mu > 0$
Construct matrix $\boldsymbol{D}$ and mapping function $F(\cdot)$
**Initialization:** $\boldsymbol{L}^0 = F(\boldsymbol{D})$, $\boldsymbol{S}^0 = \boldsymbol{E}^0 = 0, k = 0$
**while** not converge do
  $k = k + 1$
  Update $\boldsymbol{L}^k, \boldsymbol{S}^k, E^k$ via (24)
  Update $\boldsymbol{S}^k$ by performing $K$ iterations of (21)
**Output:** $\left\{\boldsymbol{L}^k, \boldsymbol{S}^k, \boldsymbol{E}^k\right\}$

---

### 2.4. Motion-Track Region Extraction

A foreground image containing only moving targets after low-rank sparse decomposition of the video SAR image sequence is expected to be obtained. In the previous section, an improved method is proposed to obtain a better decomposition effect. However, in the actual processing of complex scenes, there are some special cases that make the matrix decomposition unsatisfactory. Some clutter still exists in the foreground image, which interferes with moving target extraction. On the one hand, tracking area extraction can focus on the target more accurately by changing the detection range. For this task, local segmentation is better than global segmentation, and local detection will have more accurate detection. On the other hand, the movement trajectory can also provide the movement situation and position information of the target, which is convenient for other tasks such as tracking and route planning. There are also many ways to extract motion-trajectory areas. In this paper, we hope to find a more robust method to combat the problem that the shapes of shadows of moving targets in video SAR are variable and the gray distribution is also variable, which leads to the problem that the features are unlikely to be uniform. Therefore, an algorithm for extracting the motion trajectory area is designed for the foreground matrix obtained by matrix decomposition. In the detection process, the moving target can be extracted based on the track area while suppressing the surrounding clutter. The algorithm consists of two steps: one is to roughly extract the trajectory using double-threshold segmentation, and the other is to eliminate the wrongly extracted trajectory area. The process is shown in Figure 4. The goal of rough extraction is to preserve the integrity of the trajectory as much as possible based on the foreground matrix information, while the second step of area screening is the process of trajectory refinement, to accurately retain the correct trajectory area.

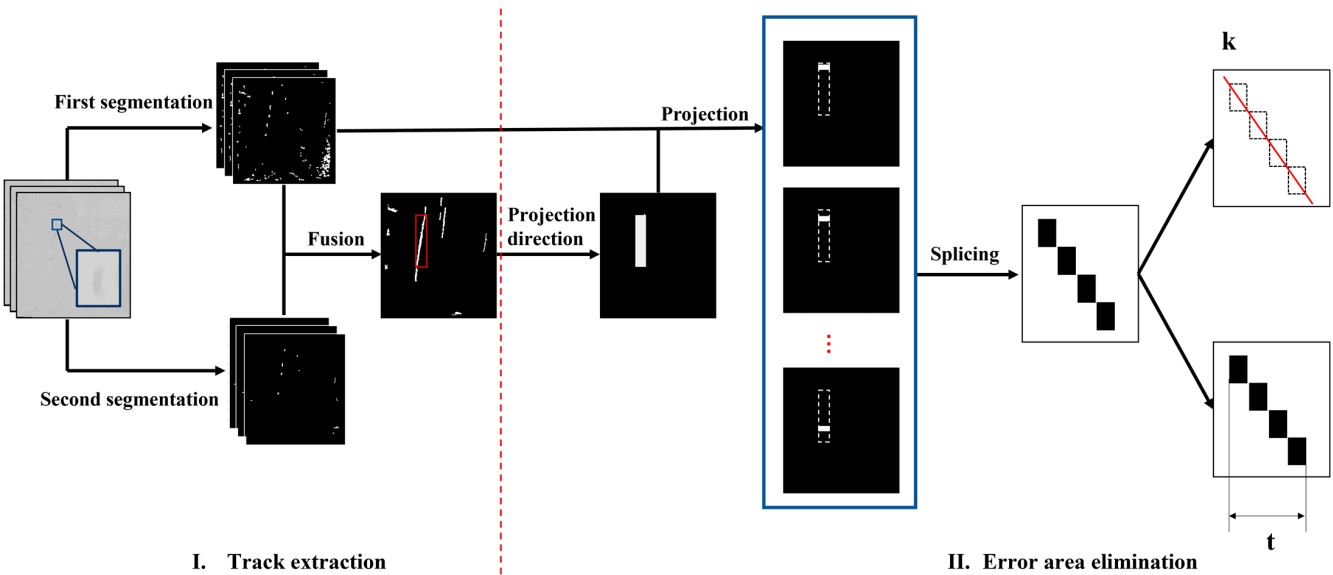

**Figure 4.** The process of motion-track region extraction.

(1) Rough extraction of the motion trajectory. For shadow detection [21], analyzes shadow characteristics. The gray value of the shadow shows a distribution that is deep in the middle and light at both ends, so setting different thresholds will lead to different segmentation results. However, the center position of the moving target is basically fixed after being segmented by different thresholds, and the clutter does not have this property. In addition, the moving target is spatially continuous, and multiple frames of images can be superimposed to construct the motion trajectory. Two difference thresholds are set to segment the multi-frame images twice, and the results of the segmentation are accumulated to form two trajectory areas. The connected area formed by the high threshold is used to ensure the integrity of the track area, and the dotted area formed by the low threshold is used to preserve the center position of the object. The center of the target generated by the low threshold is used as the basis for judging the existence of the target, the high threshold track area containing the low threshold point trace is reserved, and some wrong track areas are eliminated in this process. OTSU is used as the segmentation scheme, mainly for the binary segmentation of the background and non-background, but not for distinguishing the moving objects from the clutter.

(2) Error trajectory elimination. The initially extracted track regions are not completely accurate. To prevent the misclassified regions from affecting the detection accuracy, the wrong regions need to be eliminated. There are significant differences in the spatio-temporal characteristics of the moving targets and clutter. The spatial position of the moving target is continuous and has a longer time distribution in the image, but the clutter does not have such characteristics. On this basis, an algorithm is designed to eliminate the error trajectory area. First, the "principal direction" is calculated for each trajectory, and the smallest bounding rectangle is found for each trajectory, defining the long side of the rectangle as the "principal direction" for that trajectory. Secondly, an "and" operation is performed on the initially extracted trajectory area and the segmented foreground image. According to the "principal direction" of each trajectory, it is compressed into a one-dimensional vector, and the gray value of the part of the one-dimensional vector with the target is 1, and the values of the rest are 0. The one-dimensional vectors generated by different frames of the same trajectory are spliced frame by frame to construct the spatiotemporal trajectory state diagram of the target. Finally, linear fitting is used to calculate the slope of the state diagram, and a threshold is set to eliminate misjudged trajectory areas according to the duration of the trajectory distribution. There is a difference in the fitting slope between the moving

target and the clutter spatio-temporal state diagram, and the slope judgment threshold can be set adaptively by calculating the jump position value after statistical analysis.

## 3. Experiment

In this section, the video SAR data from the Ku-band SNL laboratory are used for verification. The scene of the data is the gate of Kirtland Air Force Base, and 100 frames were selected for detection. The image size was $660 \times 720$ pixels. Parameter $\lambda_S = 0.02$, $\lambda_E = 0.02$, and $\lambda_R = 0.04$. The difference between the two segmentation thresholds was 0.015, and the trajectory distribution duration threshold was set to three frames. The experiment was divided into two parts. The first part set up ablation experiments to prove the effectiveness of the improved LRSD model. The second part verified the effectiveness of the proposed algorithm. The evaluation index combined the qualitative analysis and quantitative analysis.

### 3.1. Performance Comparison Experiment of the Improved LRSD Model

We first demonstrate the gains from improving the LRSD model. The RPCA and TVRPCA models were set as the control groups. The decomposition results of the 21st, 42nd, and 73rd frames were for display, as shown in Figure 5. In the figure, the blue box is used to mark the location of the target, and the blue dashed box indicates that it is not easy to identify. Additionally, some strong interference clutter decomposition is marked with the red dotted circles. The shadow targets occupied fewer pixels in the scene and were not easily observed directly. Therefore, in the results of the improved model, some target shadows are enlarged and displayed to prove the accuracy of the method.

Intuitively, in the results obtained by RPCA, strong clutter scattering points appear in the three examples of images. The interference of salt-and-pepper noise is also present in the image, and its overall quality is rough. Frames 42 and 73 have obvious road-edge interference, which shows that the space-varying characteristics of radar backscatter coefficient cause the original static object background to become a dynamic background. The RPCA model retains a lot of clutter information, and the gray distribution of this clutter is similar to the target, which is not conducive to subsequent detection. In the results obtained by TVRPCA, compared with RPCA, the image quality has been improved, and the salt-and-pepper noise has basically disappeared. In frames 42 and 73, the energy of strong interference points in the foreground image decomposed by the RPCA model is weakened or has disappeared, but there are also road edges, and some new interference points are highlighted. TVRPCA enhances the recognition of the target and the background, and the total variation effect makes the image smooth, but edge clutter and strong interference still exist in the foreground image. In contrast, the foreground image obtained by the improved model proposed in this paper is flat and smooth, and the shadow can be observed intuitively. The salt-and-pepper noise in the former two and the dynamic background in the scene are further eliminated. Therefore, the decomposition effect of the model proposed in this paper is significantly improved compared with the former two. The targets in the foreground image are relatively complete, and few targets are difficult to identify due to the smoothness and less interference. For the quantitative analysis, standard deviation was used as a measure. When the target exists in the foreground image, a lower standard deviation proves that the less clutter remains in the foreground and that the decomposition effect is better. Three frames of images in Figure 5 were selected as the experimental frame, and the smallest circumscribed rectangular area containing all the targets was selected as the observation area. Then, we calculated the average standard deviation of the three frames of images. The result of the RPCA method is $1.1 \times 10^{-4}$, the result of the TVRPCA method is $1.7 \times 10^{-2}$, and the result of the improved model is $3.1 \times 10^{-3}$. A low standard deviation means less clutter in the decomposed foreground, which proves that the improved model improves the decomposition effect.

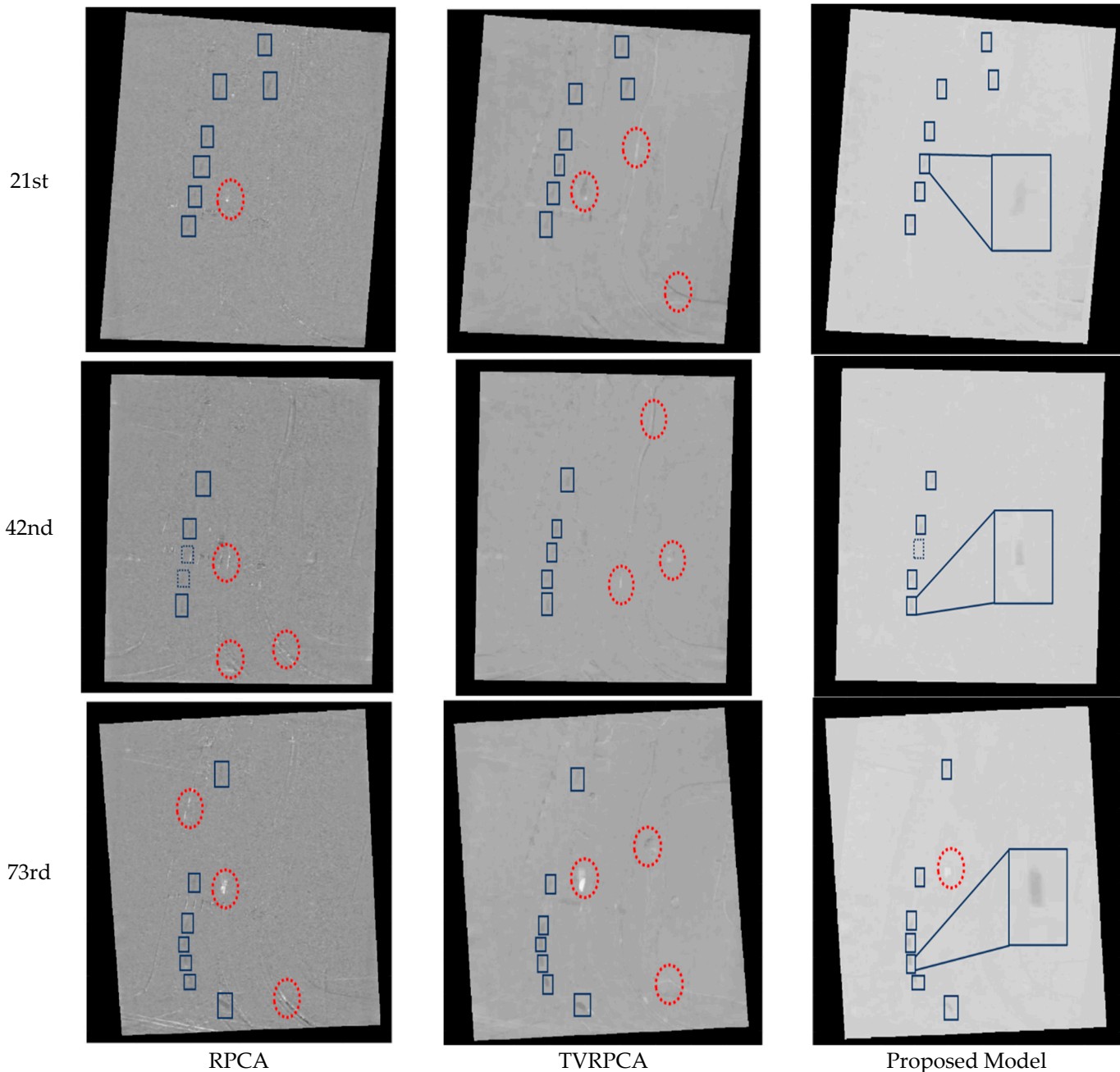

**Figure 5.** Comparison of improved models.

### 3.2. Shadow Detection Experiment

This section mainly demonstrates the detection effect of the proposed method. Figures 6–8 show the detection results of selected image frames. The displayed results include (a) the original video SAR image, (b) the foreground image generated by the improved LRSD, (c) the motion trajectory generated by using the multi-frame image with the trajectory area extraction method, and (d) the final detection result. The target shadow position is marked by a blue box in the figure. The target shadow of part of the foreground image is also enlarged to show the decomposition result more intuitively. In the experiment, the effect of directly segmenting the foreground image to extract the moving target was not good. The radar image is not as smooth and uniform as the optical image, and the gray distribution of the remaining interference points is similar to the target object, which cannot be distinguished from the moving target by segmentation, resulting in false alarms. Ex-

tracting the track area can narrow the detection range and prevent interference from other areas. Focusing on moving target area detection can also make gray-scale segmentation work well. It can be seen from the experimental results that the target detection through the matrix decomposition and the track area has excellent results. The target shadows in the figure were effectively detected, and the clutter interference was effectively removed. The shadow outline of the target is relatively intact. These indicators prove the improved performance of the algorithm.

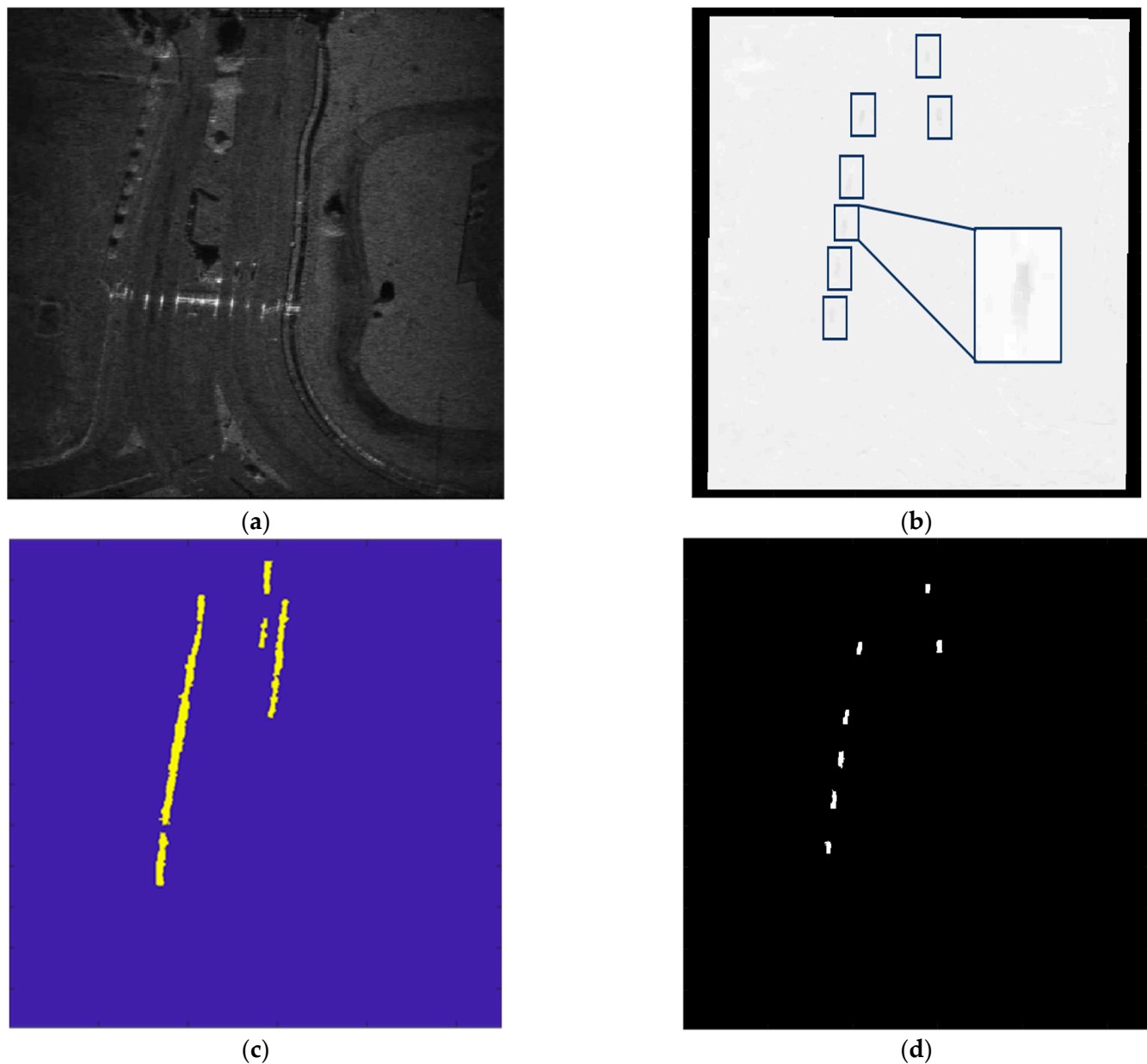

**Figure 6.** Frame 19 detection results: (**a**) video SAR image; (**b**) foreground image; (**c**) motion track; (**d**) final detection result.

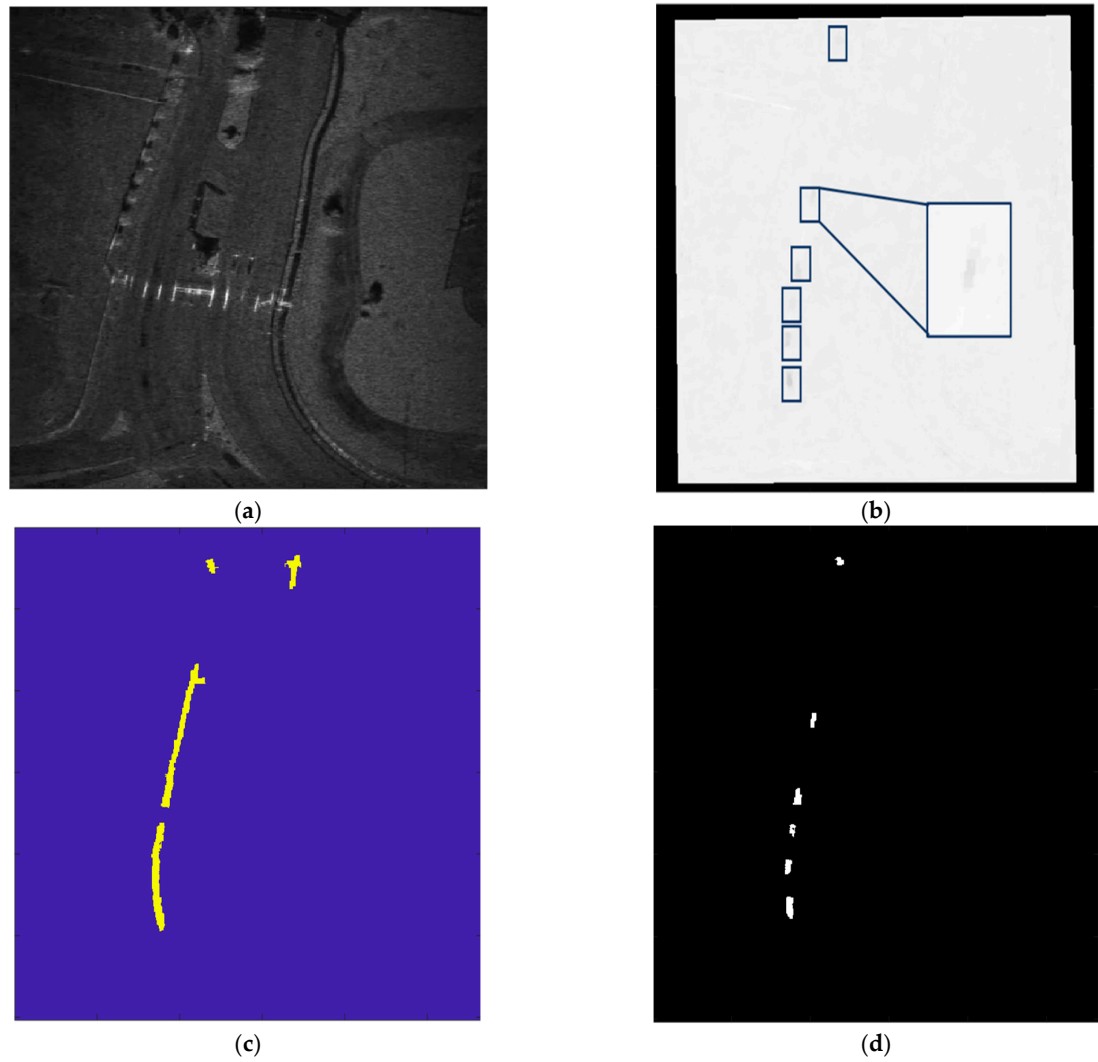

**Figure 7.** Frame 49 detection results: (**a**) video SAR image; (**b**) foreground image; (**c**) motion track; (**d**) final detection result.

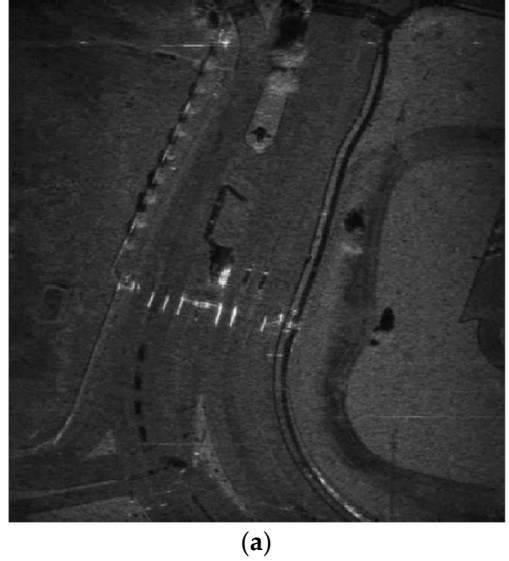

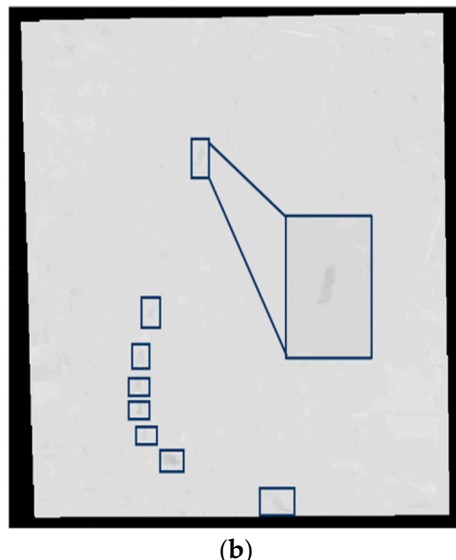

(**a**)                    (**b**)

**Figure 8.** *Cont.*

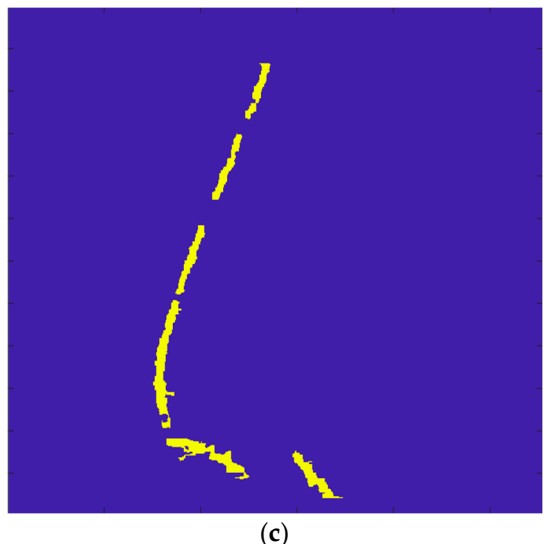
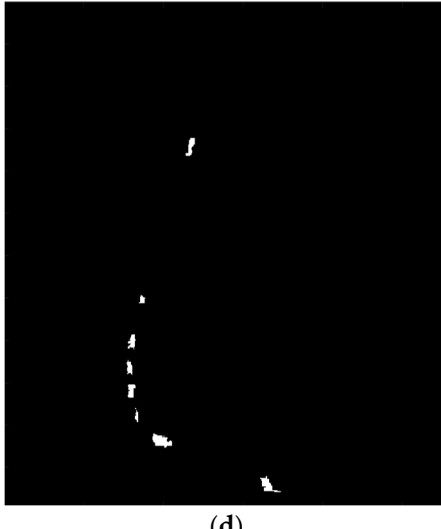

**(c)**        **(d)**

**Figure 8.** Frame 89 detection results: (**a**) video SAR image; (**b**) foreground image; (**c**) motion track; (**d**) final detection result.

In addition, five sets of comparative experiments were set up. Since there are few related studies on video SAR shadow detection using the improved RPCA model, only a few methods of the same type were introduced in this comparison experiment, and the classic methods of deep learning and background modeling were added for performance comparison. Control group 1 is the mean background modeling method. Control group 2 is a deep learning method (Faster R-CNN) [31]. Control group 3 is the RPCA [15] method. Control group 4 is the PRPCA [24] method. Control group 5 is the result of the threshold segmentation after using the improved model matrix decomposition. The quantitative analysis results are shown in Table 1 below.

**Table 1.** Comparison of results of different methods.

| Methods | Total | TP | FP | FN | P (%) | R (%) |
|---|---|---|---|---|---|---|
| Background modelling | 680 | 386 | 515 | 294 | 42.84 | 56.76 |
| Faster R-CNN [31] | 680 | 466 | 310 | 214 | 60.05 | 68.53 |
| RPCA [15] | 680 | 524 | 635 | 156 | 45.21 | 77.06 |
| PRPCA [24] | 680 | 586 | 574 | 94 | 50.52 | 86.17 |
| Improved model | 680 | 615 | 235 | 65 | 72.35 | 90.44 |
| Proposed method | 680 | 641 | 53 | 39 | **92.36** | **94.26** |

In Table 1, precision (P) and recall (R) are used to reflect the detection performance of the algorithm. True positives (TP) are the number of correct detections, and false positives (FP) are the number of false alarms. False negatives (FN) are the number of missed detections. $P = TP/(TP + FP)$ $R = TP/(TP + FN)$. The number of false alarms is the number of wrongly judged targets, and the number of missed detections is the number of missed targets. The number of correct detections, the number of false alarms, and the number of missed detections directly reflect the accuracy of target detection, and the accuracy and recall rate reflect the comprehensive detection performance of the algorithm. The higher the index, the stronger the detection performance.

It can be seen from the experimental results that the mean background modeling difference does not perform well in video SAR shadow detection, and there are a large number of false alarms. The spatial variability of the backscatter coefficient of the SAR image leads to a dynamic background, and the static background generated by the background modeling method is not completely suitable. At the same time, the background modeling is very careful in the selection of the difference threshold, so a large number of missed detections

and false alarms occur. In the Faster R-CNN method, the number of correct detections has increased compared to the former. The shadows produced by other ground objects in the video SAR image lead to misjudgment of the detection results, and the change in shadow shape and gray level will also make the network feature matching invalid and miss detection. Therefore, there are still many false alarms with this method. The RPCA method is also not robust against dynamic backgrounds. The dynamic background is sorted into the foreground matrix. Although the matrix decomposition has a certain denoising ability, there are a large number of false alarms and missed detections in the detection results after segmentation, and the effect is not good. In comparison, PRPCA improved the model and updated the sparse measure, so the matrix decomposition effect was improved, which is conducive to the detection of moving target shadows. The number of positive detections is significantly improved, and the recall rate index is good. However, there are a large number of false alarms, resulting in low detection-rate indicators. In this article, a control group of the detection method with the improved model and OTSU segmentation was also added. The new model improves the efficiency of matrix decomposition, and the target can be correctly divided into the foreground matrix, which is manifested by a higher number of positive detections. However, the dynamic interference when the gray distribution is close to the target leads to a high number of false alarms after segmentation. After the trajectory area extraction method was added, the precision and recall rates were both at a high level. Through the comparison of the above methods, the superiority of the method in this paper in video SAR moving target detection is highlighted. The improved model and optimized solution method can obtain a better matrix decomposition effect, and the extraction of the motion trajectory area can not only eliminate redundant interference but also be used to judge the motion state of the target.

## 4. Discussion

It is still possible to improve the accuracy of video SAR moving target shadow detection. The difficulties that need to be overcome come from the dynamic background and the strong interference produced by the instability of the backscatter coefficient, as well as the fluctuating background noise. At present, the matrix decomposition method applied to the video SAR shadow detection model is relatively basic. Most scholars focus on segmentation methods and use matrix decomposition as a preprocessing step, which makes detection in complex scenes inconvenient. Therefore, this paper focuses on the efficiency of the matrix factorization model. By analyzing the differences in imaging characteristics between optical images and radar images, a new attempt is made to solve many new problems in the decomposition process of video SAR images. A new decomposition framework is introduced to distinguish moving targets from dynamic backgrounds by introducing TV norms and related suppression terms. Global constraints are added for the strong interference points, and additional dynamic interference items are added to remove them. The above processing strives to obtain high-quality detection results through matrix decomposition. A comparative experiment is set up in this paper to prove the excellent performance of the improved model. Furthermore, adjustments are made to the post-processing steps. Unlike the traditional improved segmentation method, it is also an excellent processing method to narrow the detection range and focus on the target by using the motion-track-area extraction method. This method can directly eliminate all interference outside a specific area but has higher requirements in terms of the accuracy of the area. In the future, it will be necessary to integrate more space–time information into the matrix decomposition model, and the model will obtain better factorization performance. Using matrix decomposition as a preprocessing step for object detection yields high returns. In addition, the post-processing steps will also be researched and improved in order to obtain more accurate trajectory areas and adaptive parameter-setting mechanisms.

## 5. Conclusions

This paper focuses on the research of video SAR high-precision shadow detection. Firstly, the feasibility of the matrix decomposition method for video SAR moving target shadow detection is analyzed, and then a new shadow detection scheme is designed on this basis. (1) Based on the RPCA model, the total variational function is used to introduce more prior information, and a new decomposition framework is constructed to improve the effect of matrix decomposition. (2) A follow-up processing method is designed for dual-threshold trajectory segmentation and error trajectory elimination, which can focus on specific areas to achieve fast and accurate detection. Finally, the excellent performance of the method is proved using comparative experiments.

**Author Contributions:** Conceptualization, Z.Y. and M.Z.; methodology, Y.R.; funding acquisition, M.Z; software, Z.Y.; validation, Z.Y. and M.Z.; formal analysis, Y.R.; investigation, Z.Y.; resources, M.Z.; writing—original draft preparation, Z.Y.; writing—review and editing, M.Z. All authors have read and agreed to the published version of the manuscript.

**Funding:** This research was supported by Civil Aviation Project (D010206).

**Data Availability Statement:** https://www.sandia.gov/radar/pathfinder-radar-isr-and-synthetic-aperture-radar-sar-systems/video/ (accessed on 10 November 2022).

**Acknowledgments:** The authors would like to thank Huimin Zheng for his support in the early validation experiments and Yifei Liu for his help with the model solution.

**Conflicts of Interest:** The authors declare no conflict of interest.

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
