# Peer review of "A ViSAR Shadow-Detection Algorithm Based on LRSD Combined Trajectory Region Extraction"

_remotesensing, doi:10.3390/rs15061542_

Round 1

Reviewer 1 Report

In this paper, the target detection method for video SAR images is systematically discussed. To improve the background suppression effect, a low-rank sparse decomposition (LRSD) model combined with total variation (TV) regularization and coherent suppression terms, is adopted. To further improve the detection performance, the dual-threshold trajectory segmentation and error trajectory elimination methods are used jointly. This work seems to be interesting. However, the work would be excellent if the problem described below were modified and given some explanation.

1 In section 2.2, the authors mention that "Here, it is assumed that the dynamic background is sparser than the smoothed foreground, so the total variation can be used as a sparse measure of the foreground target and the dynamic background. " Why is it possible to distinguish dynamic background from foreground by TV? Please explain it in detail.

2 In Section 2.2.3, the authors say that "if M(x,y) = 1, then F(X(x,y)) = 0.' Thus, the corresponding position in the resultant figure should be shown in black. However, in Figure 3, the position corresponding to the point with the feature mask value of does not seem to exhibit the above result. Please check it.

3 To evaluate the performance of the LRSR model, the authors presented the qualitative results of LRSD in Figure 5. However, the presentation effect of the results may not be very clear. Some quantitative experiments should be added.

4 Many logical statements in the article seem to be awful, which makes the article a bit difficult to understand.

Reviewer 2 Report

In this paper, a video SAR moving target shadow detection algorithm based on low-rank sparse decomposition combined with trajectory area extraction is proposed to obtain  shadow detection results. The method includes a new lrsd decomposition framework and a trajectory based detector to improve the detection performance in dynamic scene. The method is validated by real dataset. The paper is well organized, before acceptance, there are some problems need to be solved.

1)     Section 2.2 "According to the parameter setting theory in the RPCA model, the parameter setting before the decomposition item is " lacks literature support. It is recommended to add relevant references here to support it.

2)     The TV constraint has a smoothing effect on the image. Will the shadow detection in this paper be undetectable due to smoothness loss? Please analyze the impact of TV constraints on shadow detection and discuss solutions.

3)     How complex is the improved LRSD model proposed in the paper compared with other models? Please discuss the time overhead of the algorithm in the experiment and whether the algorithm meets the real-time requirements.
